# The Pattern of RNA Editing Changes in Pleural Mesothelioma upon Epithelial-Mesenchymal Transition

**DOI:** 10.3390/ijms24032874

**Published:** 2023-02-02

**Authors:** Emanuela Felley-Bosco, Weihong Qi, Didier Jean, Clément Meiller, Hubert Rehrauer

**Affiliations:** 1Laboratory of Molecular Oncology, Department of Thoracic Surgery, University Hospital Zurich, Sternwartstrasse 14, 8091 Zürich, Switzerland; 2Functional Genomics Center Zurich, ETH Zurich and University of Zurich, 8057 Zurich, Switzerland; 3Functional Genomics of Solid Tumors, Centre de Recherche des Cordeliers, Inserm, Sorbonne Université, Université Paris Cité, F-75006 Paris, France

**Keywords:** pleural mesothelioma, RNA editing, epithelial-to-mesenchymal transition

## Abstract

Pleural mesothelioma (PM) is a cancer where epithelioid, biphasic and sarcomatoid histotypes are observed. Sarcomatoid PM is characterized by mesenchymal features. Multi-omics have been used to characterize the epithelial-to-mesenchymal (EMT) phenotype at the molecular level. We contribute to this effort by including the analysis of RNA editing. We extracted samples with the highest vs. lowest Epithelial score from two PM cohorts and observed increased RNA editing in introns and decreased RNA editing in 3′UTR upon EMT. The same was observed in primary PM primary cultures stratified by transcriptomics analysis into two groups, one of them enriched with mesenchymal features. Our data demonstrate that, as has been observed in other cancer types, RNA editing associates to EMT phenotype in PM.

## 1. Introduction

Pleural mesothelioma (PM) is the most common cancer arising from the mesothelial cell layer [1]. Traditionally, the major histologic types of mesothelioma have been the main histologic indicators of prognosis. Indeed, patients with sarcomatoid and biphasic tumors have significantly worse overall survival compared to patients with epithelioid tumors [2]. Recent studies based on multi omics approaches [3,4,5,6,7] have refined the classification into four groups or into gradients based on molecular profiles.

Transcriptome heterogeneity can be further enhanced by RNA editing. A-to-I RNA editing is the most prevalent form of RNA editing in higher eukaryotes. It is catalyzed by enzymes known as adenosine deaminases acting on double-stranded RNA (dsRNA) (ADAR). More than 85% of human primary transcripts undergo RNA editing. A large proportion of these editing sites are within repetitive elements in untranslated regions (UTR) and introns of transcripts, such as Alu elements, which have the ability to form secondary dsRNA structures [8]. ADAR activity results in the hydrolytic deamination of adenosine to form inosine, which is then interpreted as guanosine [9]. The molecular consequences of ADAR activity largely depend on the region of RNA that is targeted. For example, editing in coding sequences can change the encoded amino acid, in introns it can affect alternative splicing of transcripts, and in UTR it can alter RNA stability or the translation efficiency [10]. We have recently shown that RNA editing is increased in PM compared to mesothelial cells [11] and that RNA editing in 3′UTR varies according to *BRCA-associated protein 1 (BAP1)* status. In this study, our aim was to provide additional information on the molecular profile associated with different RNA editing patterns.

## 2. Results

### 2.1. Mesothelioma Tumors’ Transcriptome Clusters Differ in the Pattern of RNA Editing

We have recently described [11] how unsupervised clustering of the TCGA [5] and Bueno’s [3] cohorts of PM samples, based on the genomic localization of RNA editing sites, separated them into six groups, with the largest editing differences in introns: regions 5 kb downstream of genes and 3′UTR regions. RNA editing clusters 1, 2 and 3 have, on average, the highest rate of RNA editing in introns and the lowest editing rate in 5 kb downstream of gene and 3′UTR compared to clusters 4, 5 and 6 (Appendix A). We have observed a similar pattern in primary mesothelioma cultures, suggesting that the editing activity heterogeneity in tumor tissue is also present in tumor cells. To investigate the relationship between gene expression profile and RNA editing clusters, we took advantage of the recent characterization of TCGA and Bueno’s samples using the normalized epithelioid score (E-score), which has been estimated based on the transcriptome analysis, and has demonstrated that pleural mesotheliomas have different proportions of epithelioid and sarcomatoid components (E-score and S-score) [6]. Since the E-score is not normally distributed (Figure 1a), we selected in each cohort the samples with the highest and lowest E-score to explore the frequency of the occurrence of the six RNA editing clusters in each of the two groups. Patterns of RNA editing varied significantly in the two groups (Figure 1b). The group with the highest norm E-score has a higher RNA editing frequency in 3′UTR and 5 kb downstream of the gene and a lower frequency of editing in introns regions compared to the group with the lowest E-score.

Altogether, these data indicate that the genomic distribution of RNA editing events varies with epithelial to mesenchymal transition (EMT).

Of note, consistent with a low E-score being associated with a disrupted NF2/Hippo pathway [12], and the latter with ADAR2 expression [11], higher expression levels of ADAR2 were observed in low E-score samples in the TCGA dataset, but the difference was not significant in Bueno’s dataset (Appendix A).

### 2.2. Mesothelioma Primary Cultures with Different Molecular Profile and Gene Alterations Subgroups Differ in the Pattern of RNA Editing

To further support the effect of different RNA editing patterns upon EMT, we next analyzed the rate of the different RNA editing clusters in the primary mesothelioma cultures that we have robustly classified into two groups (C1 and C2) with different molecular profiles, gene alterations and histology subtypes. The epithelioid histotype is found in both groups, and tumors classified in the C2 group are associated with a worse survival prognosis. The C1 group exhibits more frequent *BAP1* alterations and the C2 group presents a mesenchymal phenotype [4]. As for the tumor transcriptome described above, the RNA editing pattern significantly changed with the same trend for a lower frequency of editing in 3′UTR and 5 Kb downstream of the gene region, with higher RNA editing in introns upon EMT (Figure 2). This confirms, using an independent phenotype classification, that RNA editing associates to EMT phenotype.

Consistent with the observation on low E-score, higher expression levels of ADAR2 were observed in C2 primary mesothelioma cultures (Appendix A).

## 3. Discussion

In this study, we provide evidence that higher levels of RNA editing in introns and lower levels of RNA editing in 3′UTR are associated with the mesenchymal pleural mesothelioma phenotype. The differential editing in these regions had been already associated with EMT in seven other cancer types [13], but had not yet been explored in PM. The differential editing had been attributed to cancer cells and we confirmed this observation in PM where differential editing was conserved in primary mesothelioma cultures.

Increased editing in introns is inversely correlated with pre-mRNA splicing [14], including in PM [11]; therefore, we put forward the hypothesis that these two RNA processes are involved in EMT characterizing the PM subgroups. The involvement of alternative splicing in development and phenotypic plasticity has been largely documented in several organisms in physiological and pathological conditions (reviewed in [15,16]). Splice variants have been identified in mesothelioma with mutated SF3B1 splicing factor [3], but have not yet been systematically analyzed, although we have documented their occurrence in several mesothelioma-relevant genes such as *lncRNA GAS5*, *CALB2* and *RBM8A* [17,18,19], or in major mesothelioma tumor suppressors such as *NF2* [20] and *BAP1* [21]. Splice variants have functional consequences, including in response to therapy. For instance, mesothelioma cells with higher levels of *BAP1*Δ splice variant are more sensitive to poly(ADP-ribose) polymerase inhibition by olaparib, indicating functional consequences of altered splicing [21].

The contribution of splice variants to EMT has been described in lung cancer (reviewed in [22]) but has not yet been explored in PM, although we have observed [11] an inverse relationship between RNA editing at splice sites and the alternative splicing in *Filamin B* (*FLNB*), which was shown to be associated with the epithelial to mesenchymal phenotype in breast cancer [23].

In breast cancer patients, different cancer subtypes can be distinguished based on exon skipping splicing features within the four main subtypes, with the luminal subtypes closely connected and the basal subtype clearly separated [24]. The subtypes of breast cancer have been suggested to arise from different populations of stem cells and progenitor cells present in the normal mammary gland [25,26]. Since we observed an inversed relationship between RNA editing and exon skipping in PM [11], we put forward the hypothesis that PM subgroups might arise from different populations of stem-like cells/progenitors present in the mesothelium. This is consistent with the observation of differential re-activation of lateral plate mesoderm differentiation in PM [27]. Studies are ongoing to address this question.

We had previously described how RNA editing in 3′UTR is higher in tumors with mutated *BAP1* [11], and in this study we observed that RNA editing in 3′UTR is more frequent in primary pleural mesothelioma cultures of the subgroup associated with epithelioid characteristics where mutation in *BAP1* is more common. The editing of specific features in 3′UTR can influence 3′UTR folding, potentially altering their interaction with RNA binding proteins. This has downstream biological consequences, as we have recently shown in PM [19], where the editing of RBM8A 3′UTR results in decreased interaction with Musashi-2 leading to increased RBM8A protein levels [19].

The implications of RNA-related processes in PM can be inferred from the observation that the DDX3X RNA helicase is mutated in some mesothelioma tumors [3,28]. However, to our knowledge it is not known yet whether it acts as a tumor suppressor or an oncogene in PM, while both these functions have been described for other cancer types (reviewed in [29]). In addition, in a genetic model of a BAP1-proficient vs. -deficient mesothelioma line, we had observed that the 191 genes, where a differential lethality for BAP1-proficient vs. deficient score ≥0.2 had been calculated, were functionally enriched with terms associated with RNA splicing and processing [30]. These processes were also found in a screen for genes involved in so-called replicative stress [31], a term describing replication forks slowing or stalling because of endogenously or exogenously derived impediments of DNA polymerases [32]. Although replicative stress response defects are associated with cancer stem-like cells and EMT [33], the way RNA processing is mechanistically involved is largely unknown.

Future studies will likely implement long-read sequencing of full-length cDNAs, including in single cells, enabling the detection of cell type specific isoforms and of aberrant splicing isoforms in cancer cells. These isoforms are occasionally translated, presented by HLA molecules, and recognized as neoantigens, as has been recently described in non-small-cell lung cancers [34]. This implies a possible contribution of RNA editing to a differential antigenic phenotype with obvious implications for immunotherapeutic approaches.

Additional validation studies may include the analysis RNA editing and alternative splicing of the epithelioid vs. sarcomatoid regions within a biphasic tumor. Furthermore, it might be of interest to investigate the contribution of RNA editing in the increased expression of *COL5A2*, *ITGAV*, and *SPARC* genes, which is correlated with mesenchymal phenotype in mesothelioma [35]. The mRNA encoded by these genes is edited according to CA editome (https://ccsm.uth.edu/CAeditome/index.html, accessed on 10 January 2023), leading to, e.g., a loss of miRNA targeting, thereby potentially contributing to increased expression levels.

In conclusion, RNA related processes, and in particular RNA editing and by consequence RNA splicing, would deserve further investigations to precisely determine their contribution in the PM phenotype.

## 4. Materials and Methods

### 4.1. RNA Editing

Mesothelioma RNA-seq reads included in the analysis were: the TCGA- Mesothelioma cohort (*n* = 87) downloaded from the NCBI database of Genotypes and Phenotypes (dbGaP) in 2019, under phs000178.v10.p8; the Malignant Pleural Mesothelioma cohort from the Bueno study (*n* = 223), downloaded from the European Genome-phenome Archive (EGA) in 2020, under EGAS00001001563 (EGAD00001001915 and EGAD00001001916).

The method for the analysis of RNA editing clusters has been recently detailed [11]. It is based on the genomic region where RNA is edited. In addition to Bueno’s and the mesothelioma TCGA datasets, the genetically characterized pleural mesothelioma primary cultures (*n* = 64) provided by Didier Jean’s team in 2022, for which RNA-Seq was performed as described in [36], was also analyzed. Briefly, RNA-seq reads were pre-processed using fastp (0.20.0). The first 6 bases at the beginning of each read were deleted to remove priming bias [37] introduced during Illumina RNA-seq library preparation. Sequencing adapters and low-quality ends (averaged quality lower than 20 in sliding windows of 4 bp, moving from 5′ to 3′ and from 3′ to 5′, respectively) were trimmed. Reads longer than 48 nt were trimmed back to 48 nt, in order to achieve uniform maximal read length across different datasets and comparable RNA editing index values. Trimmed reads with average quality above 20 and length between 18 and 48 bp were aligned to the human reference genome (Genomic Data Commons (GDC) GRCh38.d1.vd1 Reference Sequence, https://gdc.cancer.gov/about-data/gdc-data-processing/gdc-reference-files, downloaded on 12 February 2020) using STAR (2.7.8a) with 2-pass mode. PCR duplicates were marked using Picard (2.22.8). Variants in the aligned, duplicate marked RNA-seq reads were identified using GATK (v3.8.1.0) following RNA-seq best practices workflows. In detail, mapping quality reassignment, splitting spliced aligned reads into multiple supplementary alignments and clipping mismatching overhangs were performed using “SplitNCigarReads” with options “-rf ReassignOneMappingQuality -RMQF 255 -RMQT 60 -U ALLOW_N_CIGAR_READS”. Base quality recalibration was performed using “BaseRecalibrator” with dbSNP release151_GRCh38p7 downloaded in 2018 as the true variant set. Variant calling was performed using “HaplotypeCaller” with options “-dontUseSoftClippedBases -stand_call_conf 20.0”. Called variants were filtered using “VariantFiltration” with the following options: -window 35 -cluster 3 -filterName FS -filter “FS > 30.0” -filterName QD -filter “QD < 2.0”. Variants known in dbSNP, and/or in genes encoding immunoglobulins were also filtered out using SnpSift (v4.3) and bedtools (v2.29.2), respectively. Filtered variants were annotated using SnpEff (v4.3) and GDC.h38 GENCODE v22 gene annotation. Percentages of A to G changes by genomic regions in the SnpEff csv summary file were extracted for sample clustering analysis in R (v4.1). In detail, pairwise Euclidean distances among samples were computed based on percentage values of A to G changes by genomic regions with R function “dist”.

### 4.2. C1 and C2 Primary PM Cultures Clusters

To assign each primary PM sample to the molecular subtypes of the classification in two clusters (C1 and C2), a 3-gene predictor based on qRT-PCR measurements was used as previously described [4,12].

### 4.3. ADAR1/2 Gene Expression

ADAR1/2 gene expression values from aligned RNA-seq reads were computed using htseq-count (v1.99.2) with options “-a 10 -t exon -i gene_id -m intersection-nonempty”. For the stranded FunGeST dataset “-s reverse” was set, while “-s no” was used for all other non-stranded datasets. FPKM and FPKM-UQ values were computed using R (v4.1), where transcript length information was downloaded from GDC (“genecode.gene.info.v22.tsv”).

### 4.4. Statistical Analysis

Mann–Whitney and chi-square analyses were used and have been specified when used. Error bars indicate the standard error of the mean (SEM). Statistical analysis was performed using Prism 8 (Graphpad 8.0.0).

## Figures and Tables

**Figure 1 ijms-24-02874-f001:**
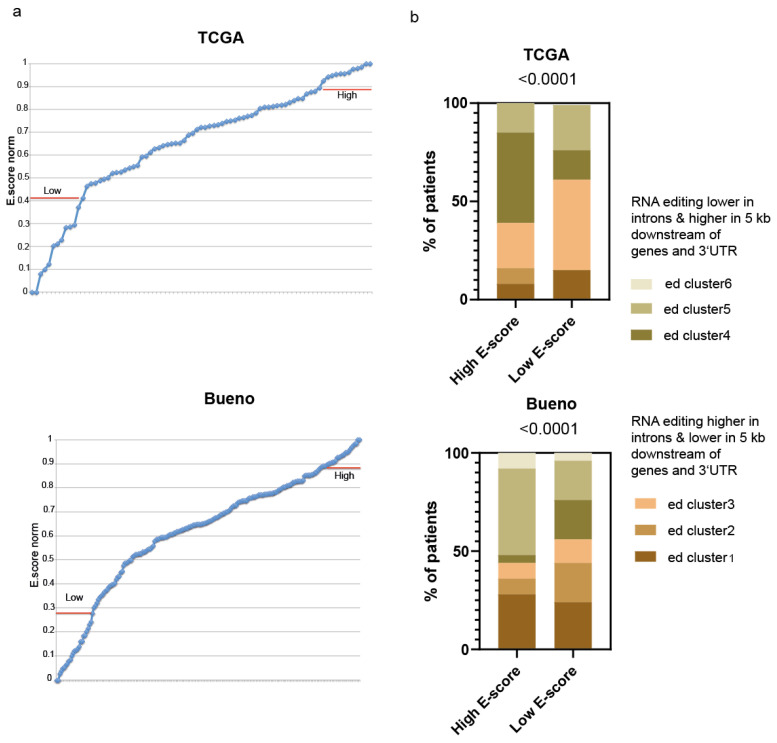
RNA editing of different genomic regions varies with epithelial-to-mesenchymal transition in pleural mesothelioma. (**a**) Norm E-score distribution in the TCGA and Bueno datasets; (**b**) Transcriptome with highest vs. lowest E-score (*n* = 13 for TCGA dataset, *n* = 25 for Bueno dataset) are characterized by differences in the genomic regions where RNA is edited. Chi-square test.

**Figure 2 ijms-24-02874-f002:**
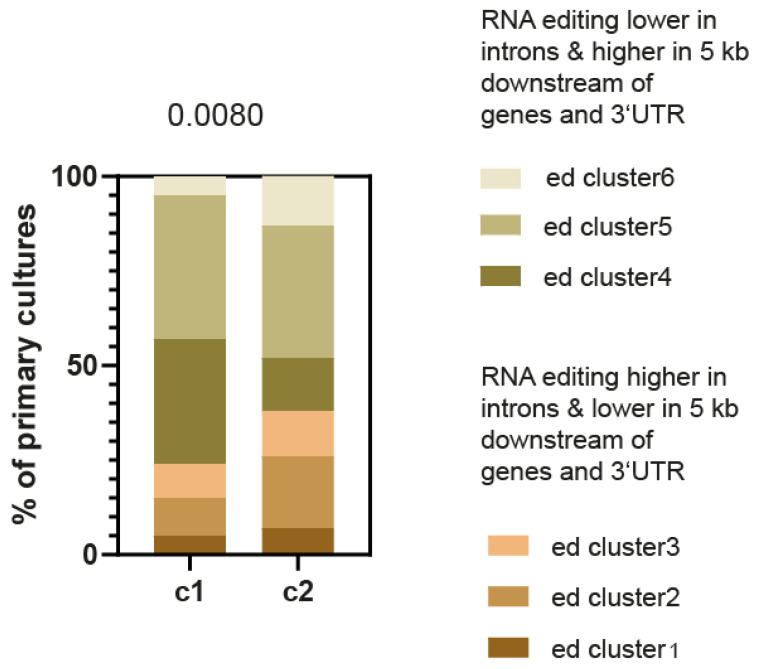
C1 (*n* = 21) and C2 (*n* = 43) primary pleural mesothelioma culture subgroups are characterized by differences in the genomic regions where RNA is edited. Chi-square test.

## Data Availability

Publicly available datasets were analyzed in this study. These data can be found at zenodo.org: 10.5281/zenodo.6504941.

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
