# Peer review of "The Pattern of RNA Editing Changes in Pleural Mesothelioma upon Epithelial-Mesenchymal Transition"

_ijms, 2023, doi:10.3390/ijms24032874_

Round 1
Reviewer 1 Report
In this short study authores analyzed changes in RNA editing clusters between pleural mesothelioma subtyptes that are classified as more epitheloid vs more sarcomatoid. As a follow up to their publication in Mol Oncol 2022 by Hariharan A. et al. they nicely demonstrate that RNA editing within intros is more abundant in the most sarcomatoid like tumors that have more severe disease outcomes, whereas 3´UTR regions and such regions that are 5kb downstream of a gene are less edited.
This seems to correlate with epithelial-to-mesenchymal transition as it is also observed in an similar fashoin in primary cell lines that are classified as more epitheloid and more mesenchymal (C1 and C2).
It would be very interesting to show, if there is a correlation of editing site selection/clusters to the expression levels of the different ADARs (Adar isoforms). Could you try to check for this from the sequencing data of the tumor samples? Further, would it be possible to perform western blots from the primary cells from C1 and C2 type, to see the protein levels of Adars?
The hypothesis, that the changes in editing clusters would affect alternative splicing in these tumors is very intriguing, especially because A to I editing and splicing have been shown to be linked in several different settings including cancer. However, if this is the case for PMs would need further proof in follow up studies.
Minor comments:
line 35: "for Adenosine Deaminases .." - redundant with line 34
line 37: ...in untranslated regions (UTR) and introns of transcripts
In all figures: for the understanding of the reader, it would help a lot to remind the reader in the figure legend or the figure label itself which of the clusters contains more of which genomic regions (introns, vs UTR and 5kb downstream).
Reviewer 2 Report
Malignant mesothelioma (MM) is an extremely malignant tumor. There are three histopathologic types of MM: epithelioid, biphasic, and sarcomatoid, with patients with non-epithelioid MM having a poorer prognosis than those with epithelioid MM. It is still unclear why MM have such different histopathologic types, and this may be extremely important in the development, diagnosis, and choice of treatment of MM.
The authors examined RNA editing using two representative genome analysis databases of MM. The results strongly suggest that RNA editing patterns differ between the epithelioid and sarcomatoid MMs and are somehow related to epithelial-mesenchymal transition (EMT) in MM. These results were also confirmed in 64 samples of primary culture cells that the authors collected. In addition, the present study confirms for the first time that EMT in MM may be associated with different patterns of RNA editing, although other types of cancers have been implicated in EMT. Although this is a very brief paper, it is a very interesting one that discusses EMT in MM from a new perspective of RNA editing.
1. One of the limitations of this study is that it is a simple comparison of different cohorts without a direct investigation of samples from the same patient. For example, since biphasic type can be observed in the same tumor, if RNA could be extracted from tumor tissues of different histopathological regions in the same case to prove the difference in RNA editing patterns, very reliable data could have been obtained. This might have been possible not only with patient tissues but also with experiments using subclones of primary cultures. It might also have been interesting to examine changes in RNA editing patterns in vitro by administering cytokines such as TGF-beta. In any case, it would be better for them to mention in the Discussion section how to validate more precisely and utilize the results of this study in the future, with including a description of the limitations of this study.
2. It would be interesting to see if there are any specific changes of genes in RNA editing patterns such as of transcription factors involved in EMT, genes related to cell morphology, or some other groups of genes. If the authors found any, they should be discussed int the Discussion section.
3. Regarding Figure 1, 13 and 25 cases were analyzed as the highest and lowest from the TCGA and Bueno data sets, respectively, but the description is not clear. At first reading, it seems as if only 13 cases were analyzed from the 87 cases in TCGA, for example. However, did they actually analyze the 13 highest and 13 lowest cases for a total of 26 cases? If so, it is difficult to understand from the original text. It would be better to add some symbols or arrows to Figure 1a to indicate the "highest" and "lowest" groups.
